# Master field equations for spherically symmetric gravitational fields beyond general relativity

Raúl Carballo-Rubio [1,2] ✉

According to general relativity, black holes are incomplete, which prevents developing a complete physical description of their dynamical formation and evolution once quantum effects are taken into account. Theories beyond general relativity may provide a more complete description of black hole interiors. In this work, the most general form of the field equations for spherically symmetric gravitational fields, in which the Einstein tensor is deformed into a conserved tensor constructed from up to second-order derivatives of the metric, is described. These equations set up the stage for the study of the dynamics of spherically symmetric spacetimes beyond general relativity, providing tools for the theoretical exploration of a paradigm of black hole physics free of the incompleteness characteristic of Einstein's theory. A general proof of the Birkhoff–Jebsen theorem for vacuum solutions, and the construction of field equations describing the effective geometrodynamics of regular black holes interacting with matter, are discussed.

The spherically symmetric Einstein field equations are a system of differential equations describing the gravitational fields of spherically symmetric distributions of matter. The study of these equations has yielded famous solutions, including the Schwarzschild[1], Vaidya[2–4], Oppenheimer–Snyder[5], and Friedmann–Lemaître–Robertson–Walker[6–12] metrics, which played a key role in the blossoming of general relativity and the consolidation of the current understanding of black holes and the large-scale properties of the universe.

In spite of the success of the spherically symmetric Einstein equations in providing the best available description of these highly symmetric situations, these equations are known to break down and therefore have a limited range of applicability. The formation of spacetime singularities[13–16] provides a strong motivation to consider physics beyond general relativity, including the open problem of combining the principles of quantum mechanics and gravity.

One of the key missing pieces in the study of physics beyond general relativity is the lack of field equations providing a cano-nical framework in which to study dynamical aspects[17]. While kinematical aspects are well understood, dynamical studies are lagging behind. As an illustrative example, it is possible to construct a purely geometric classification of all possible spherically symmetric black hole spacetimes beyond general relativity[18–21], but the dynamical laws governing the different geometric classes remain virtually unknown (a similar geometric classification exists for cosmological spacetimes[22,23]).

This article addresses this gap for spherically symmetric spacetimes, describing the structure of a set of master field equations in which the spherically symmetric Einstein tensor is deformed into an identically conserved gravitational (which, in this article, means that it is constructed solely from the metric) tensor that couples to a conserved matter source. The arguments below are guided by symmetry considerations, and the resulting equations describe a wide range of deformations of general relativity, offering unexplored avenues for the study of diverse open problems.

[1]Instituto de Astrofísica de Andalucía (IAA-CSIC), Glorieta de la Astronomía, Granada, Spain. [2]Center of Gravity, Niels Bohr Institute, Blegdamsvej 17, Copenhagen, Denmark. ✉e-mail: raul.carballorubio@iaa.csic.es

## Results

### Master field equations

The arguments in this article apply to spacetimes that can be written as warped products of the form

$$g_{\mu\nu}(y)\, \mathrm{d}y^\mu \mathrm{d}y^\nu = q_{ab}(x)\mathrm{d}x^a \mathrm{d}x^b + r^2(x)\gamma_{ij}\mathrm{d}\theta^i \mathrm{d}\theta^j. \tag{1}$$

This form is general enough to describe black holes and cosmological solutions, depending on the topology of the angular manifold. Coordinates are given by $\{y^0, y^1, y^2, y^3\} = \{x^0, x^1, \theta^2, \theta^3\}$, and $\mathrm{d}\Omega^2 = \gamma_{ij}\mathrm{d}\theta^i\mathrm{d}\theta^j$ is the angular line element. The number of spacetime dimensions has been chosen to match the most physically interesting case of 4 dimensions. However, due to the mathematical treatment being effectively 2-dimensional, working in other dimensions with a different number of angular variables is straightforward and only requires a few minor adjustments (see "Methods").

In general relativity, the dynamics of spherically symmetric spacetimes is determined by the spherically symmetric Einstein field equations. In geometrized units $c = G = 1$, these can be written as

$$G_{\mu\nu}(q, r) = 8\pi T_{\mu\nu}, \tag{2}$$

where the Einstein tensor $G_{\mu\nu}(q, r)$ is a spherically symmetric tensor, symmetric in its two indices and identically conserved, that can be derived from the action principle that results from inserting Eq. (1) into the Einstein-Hilbert action, as $G_{\mu\nu} = E_{ab}\delta_\mu^a\delta_\nu^b/r^2 - rF\gamma_{ij}\delta_\mu^i\delta_\nu^j/4$ with $E_{ab}$ and $F$ being proportional to variations with respect to $q^{ab}(x)$ and $r(x)$, respectively. In this article, the cosmological constant is implicitly incorporated into the stress-energy tensor.

Beyond general relativity, the field equations for gravitational fields are unknown. The goal of this article is deriving a set of master field equations that generalize the spherically symmetric Einstein equations to

$$\mathscr{G}_{\mu\nu}(q, r) = 8\pi T_{\mu\nu}, \tag{3}$$

where $\mathscr{G}_{\mu\nu}(q, r)$ is the most general spherically symmetric tensor, symmetric in its two indices and identically conserved, that can be derived from an action principle for the variables $q_{ab}(x)$ and $r(x)$ such that it contains up to second-order derivatives of these variables only.

### 2-dimensional Horndeski theory

The construction below leverages the structure of Horndeski theory[24,25] to describe the dynamics of warped-product spacetimes, which goes beyond its conventional interpretation. The 2-dimensional Horndeski Lagrangian can be written as:

$$\mathcal{L}_{2\mathrm{DH}} = H_2(r, \chi) - H_3(r, \chi)\Box r + H_4(r, \chi)\mathcal{R} - 2\partial_\chi H_4(r, \chi)\left[(\Box r)^2 - \nabla^a\nabla^b r \nabla_a\nabla_b r\right], \tag{4}$$

where $R$ is the Ricci scalar of a 2-dimensional metric $q_{ab}$, $\{H_i(r, \chi)\}_{i=1}^4$ are generic functions of two variables, and the convention $\chi = (\nabla r)^2$ is used.

Expressions for the variations of this Lagrangian with respect to $q^{ab}(x)$ and $r(x)$ can be obtained as particular cases of the field equations for the 4-dimensional Horndeski Lagrangian, which will not be explicitly written here but can be found, with slightly different conventions, in the literature[26]. A lengthy but straightforward calculation yields:

$$\mathcal{E}_{ab}(q, r) = \frac{1}{\sqrt{-q}}\frac{\delta\mathcal{L}_{2\mathrm{DH}}}{\delta q^{ab}} = \boldsymbol{\beta}\nabla_a\nabla_b r - q_{ab}\left(\frac{1}{2}\boldsymbol{\alpha} + \boldsymbol{\beta}\Box r\right) + \left(\partial_\chi\boldsymbol{\alpha} - \partial_r\boldsymbol{\beta}\right)\nabla_a r\nabla_b r, \tag{5}$$

$$\begin{aligned}
\mathscr{F}(q, r) = \frac{1}{\sqrt{-q}}\frac{\delta\mathcal{L}_{2\mathrm{DH}}}{\delta r} = {}&-\boldsymbol{\beta}\mathcal{R} + 2\partial_r\boldsymbol{\beta}\Box r + \partial_r\boldsymbol{\alpha} + 2\partial_\chi\boldsymbol{\beta}\left[(\Box r)^2 - \nabla_a\nabla_b r\nabla^a\nabla^b r\right] \\
&- 2\partial_r\left(\partial_\chi\boldsymbol{\alpha} - \partial_r\boldsymbol{\beta}\right)\chi - 2\left(\partial_\chi\boldsymbol{\alpha} - \partial_r\boldsymbol{\beta}\right)\Box r \\
&- 2\partial_\chi\left(\partial_\chi\boldsymbol{\alpha} - \partial_r\boldsymbol{\beta}\right)\nabla_a r\nabla^a\chi,
\end{aligned} \tag{6}$$

with the definitions

$$\boldsymbol{\alpha} = H_2 + \chi\partial_r(H_3 - 2\partial_r H_4), \quad \boldsymbol{\beta} = \chi\partial_\chi(H_3 - 2\partial_r H_4) - \partial_r H_4. \tag{7}$$

These variations satisfy the identity[24,25]

$$\nabla^a\mathcal{E}_{ab} + \frac{1}{2}\mathscr{F}\partial_b r = 0. \tag{8}$$

This can be shown explicitly by taking the divergence of $\mathcal{E}_{ab}$ and using the relations $\Box\nabla_a r - \nabla_a\Box r = \mathcal{R}\nabla_a r/2$ and $\nabla^b(\nabla r)^2\nabla_b\nabla_a r - \nabla_a(\nabla r)^2\Box r = \nabla_a r\left[\nabla_b\nabla_c r\nabla^b\nabla^c r - (\Box r)^2\right]$. Note that Eq. (8) is an off-shell statement that can be derived as a consequence of the diffeomorphism invariance of the 2-dimensional theory.

The spherically symmetric Einstein field equations are recovered for the particular functions

$$\boldsymbol{\alpha} = \boldsymbol{\alpha}_{\mathrm{GR}} = 2(1 - \chi), \quad \boldsymbol{\beta} = \boldsymbol{\beta}_{\mathrm{GR}} = -2r, \tag{9}$$

which satisfy $\partial_\chi\boldsymbol{\alpha}_{\mathrm{GR}} - \partial_r\boldsymbol{\beta}_{\mathrm{GR}} = 0$. From now on, it will be implicitly assumed that $\boldsymbol{\alpha}$ and $\boldsymbol{\beta}$ are non-trivial functions with an asymptotic behavior in the $r \to \infty$ limit matching the expressions in Eq. (9).

### Definition of a conserved gravitational tensor

The main definition in this article is the spherically symmetric tensor

$$\mathscr{G}_{\mu\nu}(q, r) = \mathscr{G}_{ab}\delta_\mu^a\delta_\nu^b + \mathscr{H}r^2\gamma_{ij}\delta_\mu^i\delta_\nu^j = \frac{1}{r^2}\mathcal{E}_{ab}\delta_\mu^a\delta_\nu^b - \frac{1}{4}r\mathscr{F}\gamma_{ij}\delta_\mu^i\delta_\nu^j. \tag{10}$$

By construction, this tensor is symmetric in its indices and contains up to second derivatives of the gravitational variables $q_{ab}(x)$ and $r(x)$.

This tensor is also identically conserved, which can be shown by calculating the divergence

$$\nabla_\mu\mathscr{G}^{\mu\nu} = \partial_\mu\mathscr{G}^{\mu\nu} + \Gamma_{\mu\rho}^\mu\mathscr{G}^{\rho\nu} + \Gamma_{\mu\rho}^\nu\mathscr{G}^{\mu\rho} = \frac{1}{\sqrt{-g}}\partial_\mu\left(\sqrt{-g}\,\mathscr{G}^{\mu\nu}\right) + \Gamma_{\mu\rho}^\nu\mathscr{G}^{\mu\rho}, \tag{11}$$

where the relation $\Gamma_{\mu\rho}^\mu = \partial_\rho\sqrt{-g}/\sqrt{-g}$ has been used. The warped form of the metric in Eq. (1) leads to some simplifications in the Christoffel symbols. In particular, all Christoffel symbols of the form $\Gamma_{ab}^i$ vanish. Hence, the divergence above is equivalent to the two following equations for the 2-dimensional and angular indices, respectively:

$$\nabla_\mu\mathscr{G}^{\mu b} = \frac{1}{\sqrt{-g}}\partial_a\left(\sqrt{-g}\,\mathscr{G}^{ab}\right) + \Gamma_{\mu\rho}^b\mathscr{G}^{\mu\rho}, \tag{12}$$

$$\nabla_\mu\mathscr{G}^{\mu k} = \frac{1}{\sqrt{-g}}\partial_i\left(\sqrt{-g}\,\mathscr{G}^{ik}\right) + \Gamma_{ij}^k\mathscr{G}^{ij}, \tag{13}$$

with the last term in the first equation mixing the 2-dimensional and angular indices.

The second equation vanishes identically for any tensor of the form $\mathscr{G}^{ij} = \mathscr{H}g^{ij} = \mathscr{H}\gamma^{ij}/r^2$, as

$$\nabla_\mu\mathscr{G}^{\mu k} = \frac{\mathscr{H}}{r^2}\left[\frac{1}{\sqrt{-\gamma}}\partial_i\left(\sqrt{-\gamma}\,\gamma^{ik}\right) + \hat{\Gamma}_{ij}^k\gamma^{ij}\right] = \frac{\mathscr{H}}{r^2}\hat{\nabla}_i\gamma^{ik} = 0, \tag{14}$$

where the hatted quantities are defined for the metric $\gamma_{ij}$.

On the other hand, using $\gamma^{ij}\Gamma^b_{ij} = -2rg^{bc}\partial_c r$, it follows that

$$\nabla_\mu \mathscr{G}^{\mu b} = \frac{1}{r^2\sqrt{-q}}\partial_a\left(\sqrt{-q}\,r^2\mathscr{G}^{ab}\right) + \Gamma^b_{cd}\mathscr{G}^{cd} + \Gamma^b_{ij}\mathscr{G}^{ij} = \frac{1}{r^2}\nabla_a\left(r^2\mathscr{G}^{ab}\right) - \frac{2\mathscr{H}}{r}g^{bc}\partial_c r. \tag{15}$$

Replacing the definitions of $\mathscr{G}_{ab}$ and $\mathscr{H}$ in Eq. (10) shows that Eq. (15) is proportional to the left-hand side of Eq. (8), thus ending the proof.

Note that the existence of the tensor defined in Eq. (10) is not in contradiction with Lovelock's theorem[27]. Lovelock's theorem applies to tensors defined without symmetry requirements, while $\mathscr{G}_{\mu\nu}(q, r)$ is defined only on the subset of spherically symmetric spacetimes, with the warped form in Eq. (1) providing extra structure that lies outside of the applicability of the theorem's assumptions. A related aspect worth studying, but out of the scope of this article, is the relation between a variational principle for warped-product spacetimes in terms of the variables $q_{ab}(x)$ and $r(x)$ and a variational principle for 4-dimensional spacetimes in terms of $g_{\mu\nu}(y)$.

## Coupling to matter and overview of assumptions

The structure of the master field equations in Eq. (3) results from coupling the gravitational tensor defined in Eq. (10) to a conserved matter source. The properties of the matter source are discussed below, together with a more detailed breakdown of the assumptions underlying the construction of Eq. (3).

As in the Einstein field equations, the matter source can be quite general, although it must be conserved for consistency. Adding a source requires considering additional fields aside from the gravitational variables $q_{ab}(x)$ and $r(x)$, denoted collectively by $\{\psi\}$, without restrictions on their number or nature. The entire set of equations is of second order if and only if $T_{\mu\nu}$ and the matter field equations do not contain derivatives of order higher than second. This includes diverse matter sources such as perfect fluids, scalar fields and electromagnetic fields, among others.

While additional scalar fields may be present in the matter sector, only the scalar field $r(x)$ is included in the gravitational sector, allowing a wider range of couplings to the 2-dimensional metric $q_{ab}(x)$. This is the minimal necessary requirement to incorporate the degrees of freedom of spherically symmetric metrics, establishing a direct correspondence with the decomposition in Eq. (1). Including additional fields in the gravitational sector, representing additional (non-metric) degrees of freedom, is out of the scope of this article, although it is an interesting extension to consider.

Another underlying assumption is the existence of an action from which $\mathscr{G}_{\mu\nu}$ can be derived. The entire action $\mathcal{S}$ leading to Eq. (3) can be separated as

$$\mathcal{S} = \int d^4y\sqrt{-g}\left(\mathcal{L}_G/\kappa + \mathcal{L}_M\right)\Big|_{g(q,r)}, \tag{16}$$

where $\mathcal{L}_G(g)\big|_{g(q,r)}$ and $\mathcal{L}_M(g, \{\psi\})\big|_{g(q,r)}$ are the gravitational and matter Lagrangian densities for the metric in Eq. (1), $\kappa = 16\pi$ in natural units, and $\mathcal{L}_G\big|_{g(q,r)} = \mathcal{L}_{2DH}(q, r)/r^2$. Non-minimal couplings to (non-derivative) curvature invariants of the 4-dimensional metric can be present in the matter Lagrangian density. By construction, only the gravitational part of the action is deformed with respect to the Einstein-Hilbert action.

Theories in which the gravitational field equations are derivable from an action principle but cannot be put in the form of Eq. (3) must therefore introduce one of the following three ingredients in the gravitational part of the action: either higher-order derivatives, additional (non-metric) degrees of freedom, or non-localities. Note that, for specific examples of these three cases not to be included in the treatment here, there must not exist field redefinitions under which the action takes the form in Eq. (16).

A natural and non-trivial extension of the treatment in this article is motivated by known results of the existence of scalar-tensor theories beyond Horndeski in which no extra (pathological) degrees of freedom propagate[28–30]. This indicates the possibility of relaxing our requirements and constructing master field equations of higher differential order without introducing non-metric degrees of freedom, by using, in particular, Degenerate Higher-Order Scalar-Tensor (DHOST) theories[28–33] particularized to 2 dimensions.

## Birkhoff–Jebsen theorem

An essential conceptual difference between the Einstein field Eq. (2) and the master field equations in Eq. (3) is that the latter are defined for families of theories (which the boldface notation aims at emphasizing) and can therefore be used to prove general results that are not theory-specific. This is illustrated below with the Birkhoff–Jebsen theorem[34–36], which holds for the master field Eq. (3) in vacuum and thus for any gravitational theory satisfying the assumptions described in the section above.

Starting from the standard form of the 2-dimensional metric

$$q_{ab}(x)\,dx^a dx^b = -e^{2\nu(t,r)}dt^2 + e^{2\lambda(t,r)}dr^2, \tag{17}$$

where the scalar field $r(x)$ is being used as one of the 2-dimensional coordinates, the vacuum master field equations are written as:

$$\mathscr{G}_{tt} = \frac{e^{2\nu}\boldsymbol{\alpha} - 2e^{2(\nu-\lambda)}\boldsymbol{\beta}\,\partial_r\lambda}{2r^2} = 0, \tag{18}$$

$$\mathscr{G}_{tr} = -\frac{\boldsymbol{\beta}\,\partial_t\lambda}{r^2} = 0, \tag{19}$$

$$\mathscr{G}_{rr} = -\frac{e^{2\lambda}\boldsymbol{\alpha} + 2\boldsymbol{\beta}\,\partial_r\nu - 2\left(\partial_\chi\boldsymbol{\alpha} - \partial_r\boldsymbol{\beta}\right)}{2r^2} = 0. \tag{20}$$

The angular equations have not been written explicitly due to not providing additional information in vacuum. It is straightforward to check that the spherically symmetric vacuum Einstein field equations for metrics with the 2-dimensional sector in Eq. (17) are recovered for the particular functional forms in Eq. (9).

Noting that $\boldsymbol{\beta}$ is a non-trivial function, Eq. (19) implies that $\partial_t\lambda = 0$. Together with the relation $\chi = e^{-2\lambda(t,r)}$ valid for the metric in Eq. (17), it follows that both functions $\boldsymbol{\alpha}\left[r, e^{-2\lambda(r)}\right]$ and $\boldsymbol{\beta}\left[r, e^{-2\lambda(r)}\right]$, as well as their derivatives, do not have any dependence on the time coordinate $t$. Eq. (18) thus becomes an ordinary differential equation for $\lambda(r)$, while Eq. (20) implies the factorization $\nu(r) = \nu_r(r) + \nu_t(t)$. The residual time dependence in $\nu_t(t)$ can be absorbed in a redefinition of the time coordinate $t$. A linear combination of Eqs. (18) and (20) permits us to write $\boldsymbol{\beta}\partial_r(\lambda + \nu_r) = \partial_\chi\boldsymbol{\alpha} - \partial_r\boldsymbol{\beta}$, which uniquely determines $\nu_r(r)$ as a function of $\lambda(r)$ (the resulting constant of integration can also be absorbed in a redefinition of the time coordinate). Hence, the solutions of the vacuum master field Eq. (18–20) form a uniparametric family of static spacetimes, thus ending the proof. A similar mathematical statement was studied for a system of differential equations that was conjectured to be the most general second-order equations for $q_{ab}(x)$ and $r(x)$[37], and that can be shown to be equivalent to the vanishing of Eqs. (5, 6) once a redundancy is cleared up (this equivalence can also be established at the level of the action[38]).

Hence, it is not possible to violate the Birkhoff–Jebsen theorem without necessarily introducing either higher-order derivatives in the gravitational Lagrangian, additional (non-metric) gravitational degrees of freedom, or gravitational non-localities. This statement agrees with the existing literature on the applicability and limits of validity of the Birkhoff–Jebsen theorem beyond general relativity. Previous discussions in the framework of scalar-tensor and related theories provide

clarifying examples, with violations of the theorem occurring for metric $f(R)$ theories due to the field equations containing higher-order derivatives of the metric[39–41], and for scalar-tensor theories whenever the additional scalar field is time dependent[42]. Also, the straightforward extension of the statement above to higher dimensions includes the results that the Birkhoff–Jebsen theorem holds for all second-order metric theories of gravity[43,44] and for certain classes of higher-derivative theories[45,46].

### Effective geometrodynamics of regular black holes

The master field equations provide a systematic approach for the study of the dynamics of black holes beyond general relativity. It is widely expected that quantum gravity effects will have distinctive imprints on the structure of black holes, which have been partially studied within diverse quantum gravity frameworks (see, e.g., recent reviews[47–49]). Instead of committing to a specific framework, the master field equations facilitate the study of the modifications of the structure of black holes in an agnostic way, capturing all the modifications that are compatible with the underlying assumptions. This approach is similar in spirit to effective field theory and has as one of its main features the use of lower-dimensional field theories to describe the effective dynamics of higher-dimensional spacetimes with a high degree of symmetry[37].

Due to the Birkhoff–Jebsen theorem, vacuum solutions of the master field equations constitute classes of spacetimes labeled by the functions $\boldsymbol{\alpha}$ and $\boldsymbol{\beta}$. An explicit algorithm to find solutions of Eqs. (18–20) for $\nu(r)$ and $\lambda(r)$ is the following. First, integrate Eq. (18) to obtain $\lambda(r)$:

$$\frac{\mathrm{d}\left[e^{-2\lambda(r)}\right]}{\mathrm{d}r} = -\frac{\boldsymbol{\alpha}\left[r, e^{-2\lambda(r)}\right]}{\boldsymbol{\beta}\left[r, e^{-2\lambda(r)}\right]}, \tag{21}$$

from which the Arnowitt-Deser-Misner (ADM) mass arises as an integration constant. Then obtain $\nu(r)$ through the relation

$$\nu(r) = -\lambda(r) + \int \mathrm{d}r \frac{\left(\partial_\chi \boldsymbol{\alpha} - \partial_r \boldsymbol{\beta}\right)}{\boldsymbol{\beta}}\Bigg|_{\chi = e^{-2\lambda(r)}}. \tag{22}$$

As a sanity check, it is straightforward to recover the Schwarzschild solution for the choices in Eq. (9).

The correspondence between vacuum solutions and the functions $\boldsymbol{\alpha}$ and $\boldsymbol{\beta}$ goes both ways, as Eqs. (21, 22) can be solved for $\boldsymbol{\alpha}$ and $\boldsymbol{\beta}$ starting from a specific static metric. This is particularly simple in the cases for which $\nu + \lambda = 0$, which include the Schwarzschild solution as well as virtually any of the most common regular black hole metrics in the literature. Then, Eq. (22) is solved by a potential (or mass[37]) function $\boldsymbol{\Omega}(r, \chi)$ that generates $\boldsymbol{\alpha} = \partial_r \boldsymbol{\Omega}$ and $\boldsymbol{\beta} = \partial_\chi \boldsymbol{\Omega}$ off-shell, while a direct rearrangement of Eq. (21) indicates that this potential must reduce to an integration constant (the ADM mass) on-shell. Fixing an arbitrary normalization factor allows us to write

$$\boldsymbol{\Omega}(r, \chi)\big|_{\chi = e^{-2\lambda}} = 4M. \tag{23}$$

This relation can be used to define an algorithm to find the potential function $\boldsymbol{\Omega}$ algebraically from the expression of $\lambda$ in terms of the ADM mass, taking into account that $e^{-2\lambda} = g_{rr}^{-1} = \chi$ in the coordinates in Eq. (17), and using Eq. (23) to replace $M$ with $\boldsymbol{\Omega}(r, \chi)\big|_{\chi = e^{-2\lambda}}$. The resulting expression provides an implicit relation between $\boldsymbol{\Omega}(r, \chi)$ and $\chi$ valid in these coordinates. However, these quantities being scalars, this relation must hold in any coordinate system. This implicit relation must be solved algebraically to find $\boldsymbol{\Omega}(r, \chi)$.

Two of the most well-known regular black hole models, proposed by Bardeen[50] and Hayward[51], are characterized, respectively, by the

metric functions

$$e^{2\nu_\mathrm{B}(r)} = e^{-2\lambda_\mathrm{B}(r)} = 1 - \frac{2r^2 M}{\left(r^2 + \ell^2\right)^{3/2}}, \quad e^{2\nu_\mathrm{H}(r)} = e^{-2\lambda_\mathrm{B}(r)} = 1 - \frac{2r^2 M}{r^3 + 2\ell^2 M},$$

$$\tag{24}$$

with $\ell$ a new length scale. Applying the algorithm above results in the potential functions

$$\boldsymbol{\Omega}_\mathrm{B}(r, \chi) = \frac{2(1-\chi)\left(r^2 + \ell^2\right)^{3/2}}{r^2}, \quad \boldsymbol{\Omega}_\mathrm{H}(r, \chi) = \frac{2(1-\chi)r^3}{r^2 - \ell^2(1-\chi)}. \tag{25}$$

From these potential functions, it is straightforward to derive $\boldsymbol{\alpha}$ and $\boldsymbol{\beta}$ and write down specific instances of Eq. (3) for which vacuum solutions are given by the Bardeen and Hayward metrics, respectively, instead of the Schwarzschild metric. The metrics in Eq. (24) thus describe the spacetime geometry outside of localized spherically distributions of matter for the theories defined by the potential functions in Eq. (25).

A complete description must also include matter fields. For a given pair of functions $\boldsymbol{\alpha}$ and $\boldsymbol{\beta}$ (for example, the ones for the Bardeen or Hayward examples), the dynamical evolution of the gravitational variables in the region inside matter is described by the master field Eq. (3) with a non-zero matter source with stress-energy tensor $T_{\mu\nu}$. These equations provide a framework in which to study a wide array of processes, from gravitational collapse to the backreaction of classical and semiclassical perturbations, as well as possible instabilities. All these are key open issues in the study of regular black holes, in particular due to the lack of a suitable mathematical framework in which to perform the necessary calculations[17], which the master field equations described here provide.

It has been shown that strong hyperbolicity can fail in 4-dimensional Horndeski theory[26,52], and that phenomena such as the development of shocks can affect well-posedness[53]. Hence, any deformation of the spherically symmetric Einstein-Hilbert action described by 2-dimensional Horndeski theory, including the theories arising for the Bardeen and Hayward examples above, must be carefully assessed from this perspective. It is also necessary to impose additional conditions on the 2-dimensional degrees of freedom $q_{ab}(x)$ and $r(x)$ to guarantee that a meaningful 4-dimensional spacetime can be reconstructed. Without such conditions, a 4-dimensional manifold equipped with the metric in Eq. (1) will generally be geodesically incomplete, for instance due to $r(x)$ having a positive lower bound, or due to taking negative values (possibly in the maximal extension of the 2-dimensional manifold, as it has been discussed for static metrics in[54]). From the perspective of the 2-dimensional manifold, it is necessary to impose that $r(x)$ vanishes on a 1-dimensional submanifold to guarantee that the 4-dimensional manifold has a center of symmetry. Geodesic completeness of the 4-dimensional spacetime around the center of symmetry requires that any 2-dimensional tensor displays a specific behavior in an open neighborhood of this 1-dimensional submanifold, which can be obtained explicitly by demanding the local existence of a Cartesian coordinate system around the center. These conditions depend on the symmetries of the problem and hold regardless of the dynamics. Existing treatments of similar situations in general relativity, in particular axisymmetric situations[55], provide valuable guidance for numerical implementations of these consistency conditions.

## Discussion

Due to their simpler structure compared to the full set of Einstein field equations, the spherically symmetric Einstein field equations were pivotal for the development of gravitational physics beyond Newton's theory. This article puts forward an approach for the theoretical study of spherically symmetric gravitational fields beyond general relativity,

based on a set of master field equations that generalize the spherically symmetric Einstein field equations.

In this approach, a lower-dimensional field theory (2-dimensional Horndeski theory) is used to describe the effective geometrodynamics of higher-dimensional spacetimes with a high degree of symmetry (spherical symmetry). The framework and calculational tools presented here offer the possibility of identifying robust features shared by high-energy modifications of the gravitational dynamics for which the underlying assumptions are satisfied, either at a fundamental or an emergent level. It would be interesting to study whether similar sets of master field equations could be defined for situations with less (or different) symmetries.

The potential of this approach for the agnostic study of the dynamics of regular gravitational collapse in spherical symmetry has been illustrated with the discussion of two aspects: a general proof of the Birkhoff–Jebsen theorem under the same assumptions on which the construction of the master field equations rests, and the identification of specific instances of these equations describing the interaction with matter of regular deformations of the Schwarzschild black hole. Realizing this potential requires a thorough study of the interplay between these regular solutions and different matter contents that can be inserted on the right-hand side of the master field equations, and solving several open issues, such as evaluating well-posedness and implementing consistency conditions in numerical explorations of dynamical processes.

## Methods

### Expressions for different spacetime dimensions
For completeness, the generalization of Eq. (10) for $D$ dimensions (that is, $D-2$ angular variables) is provided:

$$\mathcal{G}^{(D)}_{\mu\nu}(q,r) = \frac{1}{r^{D-2}}\mathcal{E}_{ab}\delta^a_\mu\delta^b_\nu - \frac{r^{5-D}}{2(D-2)}\mathcal{F}\gamma_{ij}\delta^i_\mu\delta^j_\nu. \quad (26)$$

The proof that the tensor in Eq. (26) is identically conserved follows the same steps as the proof for the 4-dimensional tensor in Eq. (10).

On the other hand, the functional forms for $\boldsymbol{\alpha}$ and $\boldsymbol{\beta}$ for which the spherically symmetric Einstein field equations are recovered become, instead of the expressions in Eq. (9),

$$\boldsymbol{\alpha}^{(D)}_{\mathrm{GR}} = (D-2)(D-3)r^{D-4}(1-\chi), \quad \boldsymbol{\beta}^{(D)}_{\mathrm{GR}} = -(D-2)r^{D-3}. \quad (27)$$

## Data availability
Data sharing is not applicable to this article as no datasets were generated or analysed during the current study.

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

## Acknowledgments

R.C.R. is grateful for related discussions with Julio Arrechea, Carlos Barceló, Jose Beltrán Jiménez, Valentin Boyanov and Astrid Eichhorn, the financial support received from the Spanish Government through the Ramón y Cajal program (contract RYC2023-045894-I), and the Grant No. PID2023-149018NB-C43 funded by MCIN/AEI/10.13039/501100011033, and the Severo Ochoa grant CEX2021-001131-S funded by MCIN/AEI/10.13039/501100011033, and the hospitality of the Center of Gravity, a Center of Excellence funded by the Danish National Research Foundation under grant No. 184.

## Author contributions

R.C.R. conceived the concept, developed the theory, performed the calculations and prepared the manuscript.

## Competing interests

The author declares no competing interests.
