## [Transparent Peer Review file · Nature Communications]

Master field equations for spherically symmetric gravitational fields beyond general relativity

Corresponding Author: Professor Raúl Carballo-Rubio

Version 0:

Reviewer comments:

Reviewer #1

(Remarks to the Author)

The author has done a remarkably good job in addressing all the reviewers' comments and in improving the manuscript. The resubmission includes two new sections: Sec. IV clears all questions regarding the assumptions, applicability and scope of the formalism; Sec. VI outlines and exemplifies a procedure for concrete applications, and furthermore discusses potential situations where the formalism may break down. The outcome is a very solid paper, in my opinion.

My only observation concerns the lack of references on the Birkhoff-Jensen theorem in scalar-tensor theories. There exists a significant literature on the subject, including examples that satisfy and violate the theorem. I would suggest the author to add a few sentences (with relevant references) on how the presented theorem generalizes existing results stating the validity of the Birkhoff-Jensen theorem, and how known counter-examples violate some of the assumptions (for example gr-qc/0607096).

I would recommend publication provided the above comment is suitably addressed.

Reviewer #2

(Remarks to the Author)

I thank the author for having carefully revised the manuscript in response to the previous comments by the referees. I have read the new version and I believe that the manuscript has indeed been improved. In particular, the author now shows more clearly the potential applications of the derived master equation, the key point of this work.

My overall impression, however, is still that this work remains outside of the scope of a journal with the standards of Nature Communications, which primarily focuses on very high impact research with strong connections to experiments or observations (or at least with clear potential applications in that direction). My opinion is that this work, while mathematically sound and with clear potential applications, does not quite meet these criteria.

This work is still focused on beyond-GR theories, which remain a niche area without sufficiently strong motivation for the broader physics community, so the paper would be better suited for a specialized journal focused on modified gravity theories.

I have revised the paper to address the minor comments from reviewer #1, as requested. Please find more details below.

I am sincerely grateful to all reviewers for their time and commitment to refereeing over the different rounds of review and for their essential contributions to the improvement of the manuscript.

Reviewer #1

The author has done a remarkably good job in addressing all the reviewers' comments and in improving the manuscript. The resubmission includes two new sections: Sec. IV clears all questions regarding the assumptions, applicability and scope of the formalism; Sec. VI outlines and exemplifies a procedure for concrete applications, and furthermore discusses potential situations where the formalism may break down. The outcome is a very solid paper, in my opinion.

My only observation concerns the lack of references on the Birkhoff-Jensen theorem in scalar-tensor theories. There exists a significant literature on the subject, including examples that satisfy and violate the theorem. I would suggest the author to add a few sentences (with relevant references) on how the presented theorem generalizes existing results stating the validity of the Birkhoff-Jensen theorem, and how known counter-examples violate some of the assumptions (for example gr-qc/0607096).

I would recommend publication provided the above comment is suitably addressed.

I am grateful to the referee for the positive feedback regarding the previous revision, and I am glad to hear that the content of the new sections has strengthened the manuscript.

The suggestion of adding a few sentences and references including examples that satisfy and violate the Birkhoff-Jensen theorem in scalar-tensor and related theories is appreciated, and I have modified the manuscript accordingly.

To implement the reviewer's recommendation, I have added a few sentences at the end of the section discussing the Birkhoff-Jensen theorem (8 new lines in total, that can be found at the end of page 8 and beginning of page 9). These sentences offer examples of how the presented statement is compatible with the existing literature on the subject. The reference pointed out by the referee has been included, together with seven other references. The situation in higher-dimensional theories has also been briefly mentioned for completeness.